# KMU-1170, a Novel Multi-Protein Kinase Inhibitor, Suppresses Inflammatory Signal Transduction in THP-1 Cells and Human Osteoarthritic Fibroblast-Like Synoviocytes by Suppressing Activation of NF-κB and NLRP3 Inflammasome Signaling Pathway

**DOI:** 10.3390/ijms22031194

**Published:** 2021-01-26

**Authors:** Hye Suk Baek, Victor Sukbong Hong, Sang Hyon Kim, Jinho Lee, Shin Kim

**Affiliations:** 1Department of Immunology, School of Medicine, Keimyung University, Daegu 42601, Korea; thortiw@naver.com; 2Department of Chemistry, Keimyung University, Daegu 42601, Korea; victorh@kmu.ac.kr; 3Division of Rheumatology, Department of Internal Medicine, School of Medicine, Keimyung University, Daegu 42601, Korea; mdkim9111@dsmc.or.kr; 4Institute of Medical Science, Keimyung University, Daegu 42601, Korea

**Keywords:** KMU-1170, protein kinase, lipopolysaccharide, inflammation, NF-κB, NLRP3 inflammasome, osteoarthritis, fibroblast-like synoviocyte

## Abstract

Protein kinases regulate protein phosphorylation, which are involved in fundamental cellular processes such as inflammatory response. In this study, we discovered a novel multi-protein kinase inhibitor, KMU-1170, a derivative of indolin-2-one, and investigated the mechanisms of its inflammation-inhibiting signaling in both THP-1 cells and human osteoarthritic fibroblast-like synoviocytes (FLS). We demonstrated that in THP-1 cells, KMU-1170 inhibited lipopolysaccharide (LPS)-induced upregulation of inducible nitric oxide synthase (iNOS) and cyclooxygenase-2 (COX-2), and, furthermore, suppressed LPS-induced phosphorylation of transforming growth factor-β-activated kinase 1, JNK, ERK, inhibitor of NF-κB kinase α/β (IKKα/β), and NF-κB p65 as well as nuclear translocation of NF-κB p65. Moreover, KMU-1170 suppressed LPS-induced upregulation of proinflammatory cytokines such as IL-1β, TNF-α, and IL-6, and, notably, inhibited LPS-induced upregulation of the NLRP3 inflammasome in THP-1 cells. Importantly, KMU-1170 attenuated LPS-mediated inflammatory responses in human osteoarthritic FLS, such as the upregulation of IL-1β, TNF-α, IL-6, iNOS, and COX-2 and the phosphorylation of IKKα/β and NF-κB p65. Collectively, these results suggest that KMU-1170 inhibits inflammatory signal transduction and could be developed as a potential anti-inflammatory agent.

## 1. Introduction

Inflammation is a crucial and fundamental immune system response in the body that protects the host from harmful stimuli and maintains tissue homeostasis [1,2]. Although an effective inflammatory response allows survival in the case of infection or injury, excessive or sustained inflammation can lead to diverse pathological conditions, such as asthma, cancer, chronic pain, gout, mental health disorders, nervous breakdown, psoriasis, rheumatoid arthritis, and vasculitis [3]. Moreover, it is equally important that the inflammatory response to a noxious stimulus is terminated when the stimulus is removed [3]. Therefore, the balance between an effective inflammatory response and the anti-inflammatory response is one of the most important conditions for maintaining a healthy state.

Toll-like receptors (TLRs) constitute a family of critical receptors in the innate immune system that recognize various pathogens and damaged tissues [4]. TLRs are divided into several types according to their specific characteristics, such as structure, subcellular localization, and recognized molecular properties. Lipopolysaccharide (LPS), a bacterial endotoxin that is a component of the outer wall of gram-negative bacteria, is recognized by TLR4 [5]. When LPS binds to TLR4, inflammation-related gene expression is regulated through the transcription factor NF-κB [6]. Inflammation is controlled by water-soluble immune signaling molecules called cytokines [7], and inflammatory cytokines such as interleukin (IL)-1β, IL-6, and tumor necrosis factor-α (TNF-α) are induced by LPS [8]. These cytokines are involved in various inflammatory diseases and inhibiting them can potently help treat inflammatory diseases.

Inflammasome, a complex of inflammatory molecules, is an innate immune response to pathogenic infections and plays an important role in the formation of proinflammatory cytokine [9]. Among them, the NOD-, LRR- and pyrin domain-containing protein 3 (NLRP3) inflammasome function has been studied a lot and it is composed of essential regulatory proteins NLRP3, the adaptor molecule apoptosis-associated speck-like protein containing a CARD (ASC), procaspase-1, and NIMA-related kinase 7 [10]. After the transcription and translation of NLRP3 and IL-1β by NF-κB, active caspase-1 can process pro-IL-1β into its mature secreted form [11]. Therefore, it is very important to effectively regulate NLRP3 inflammasome-related signaling in inflammatory diseases.

Protein kinase is responsible for phosphorylation of protein, which plays a critical role in intracellular signaling of inflammation [12]. Protein kinases are very important drug targets in drug development for the treatment of inflammatory diseases [13]. For example, tofacitinib, a potent Janus-associated kinase (JAK) 1 and JAK3 antagonist developed by Pfizer, reduced signs and symptoms of rheumatoid arthritis (RA) and improved physical function of patients with RA [14]. Moreover, semapimod, a p38 kinase inhibitor improved clinical outcome in patient with Crohn’s disease [15]. A spleen tyrosine kinase inhibitor BAY 61-3606 was approved by the US Food and Drug Administration (FDA) of indication for patients with allergic asthma [16]. Additionally, various Bruton’s tyrosine kinase (BTK) inhibitors are in clinical trials [17]. However, promising protein kinase inhibitors still need to be developed.

In this study, we developed a novel multi-protein kinase inhibitor, KMU-1170, a derivative of indolin-2-one, and evaluated the anti-inflammatory effect of KMU-1170 in the TLR4-NF-κB-NLRP3 signaling pathway by using the human monocyte cell line THP-1 and primary human osteoarthritic fibroblast-like synoviocytes (FLS).

## 2. Results

### 2.1. Effect of KMU-1170 on the Activity of Various Protein Kinases

KMU-1170 was synthesized from 5-bromoindoline-2,3-dione as follows (Figure 1A): Compound **1** was obtained through reduction and borylation, and Suzuki coupling reaction of compound **1** with a pyrazine derivative, compound **2**, yielded compound **3**. Lastly, Knoevenagel condensation with 1*H*-imidazole-4-carbaldehyde generated KMU-1170. The kinase assay results showed that KMU-1170 inhibited numerous kinases associated with inflammatory reactions, with particularly strong inhibition for mitogen-activated protein kinase 1 (MAPK1), Lck, TYK2, JAK3, and Txk (Table 1).

### 2.2. KMU-1170 Reduces LPS-Stimulated Upregulation of Inducible Nitric Oxide Synthase (iNOS) and Cyclooxygenase-2 (COX-2) in THP-1 Cells

To investigate the toxicity of KMU-1170 in THP-1 cells, 2,3-bis-(2-methoxy-4-nitro-5-sulfophenyl)-2*H*-tetrazolium-5-carboxanilide (XTT) assay was used. The result revealed that KMU-1170 exhibited no toxicity up a concentration of 1 μM (Figure 1B). Notably, pretreatment of cells with 1 μM KMU-1170 inhibited LPS-induced upregulation of iNOS mRNA and protein (Figure 1C–E), and the pretreatment also inhibited LPS-induced upregulation of COX-2 mRNA and protein but not COX-1 mRNA/protein (Figure 1F–I).

### 2.3. KMU-1170 Inhibits LPS-Induced Proinflammatory Cytokine Production in THP-1 Cells

To evaluate the anti-inflammatory mechanism of action of KMU-1170, we examined the regulation of the levels of proinflammatory cytokines such as IL-1β, TNF-α, and IL-6. Pretreatment of cells with 1 μM KMU-1170 inhibited LPS-induced upregulation of IL-1β, TNF-α, and IL-6 (Figure 2).

### 2.4. KMU-1170 Suppresses LPS-Induced Phosphorylation of TAK1 and MAPKs in THP-1 Cells

Next, we investigated whether KMU-1170 affects TLR4 and MyD88 expression. Pretreatment of THP-1 cells with 1 μM KMU-1170 did not attenuate LPS-induced upregulation of TLR4 and MyD88 (Figure 3A,B). To identify the potential molecular targets of KMU-1170 in inflammation, we examined the phosphorylation of transforming growth factor-β-activated kinase 1 (TAK1), which regulates MAPKs and the function of NF-κB. Pretreatment of KMU-1170 inhibited LPS-induced phosphorylation of TAK1 in THP-1 cells (Figure 3C,D). Among MAPKs, only ERK and JNK were phosphorylated in response to LPS, and pretreatment with 1 μM KMU-1170 suppressed LPS-induced phosphorylation of ERK and JNK in THP-1 cells (Figure 3E,F).

### 2.5. KMU-1170 Inhibits LPS-Induced Activation of NF-κB in THP-1 Cells

LPS induced the phosphorylation of inhibitor of NF-κB kinase α/β (IKKα/β) and NF-κB p65 in THP-1 cells, but pretreatment with KMU-1170 inhibited their LPS-induced phosphorylation (Figure 4A,B). We also investigated whether KMU-1170 affects the nuclear translocation of NF-κB p65 by examining cells using fluorescence microscopy, which revealed that KMU-1170 pretreatment suppressed LPS-induced nuclear translocation of NF-κB p65 in THP-1 cells (Figure 4C).

### 2.6. KMU-1170 Attenuates LPS-Induced Activation of NLRP3 in THP-1 Cells

We next investigated whether KMU-1170 affects NLRP3 inflammasome signaling. Pretreatment of THP-1 cells with 1 μM KMU-1170 inhibited LPS-induced cytosolic release of pro-IL-1β and IL-1β (Figure 5A). Moreover, pretreatment with 1 μM KMU-1170 attenuated the upregulation of NLRP3, ASC, pro-caspase-1, and pro-IL-1β induced by LPS and by co-treatment with LPS and ATP (Figure 5A). Our quantified results showed that KMU-1170 significantly inhibited NLRP3 inflammasome activity (Figure 5B).

### 2.7. KMU-1170 Inhibits LPS-Induced Inflammation in THP-1 Cells

Considering the action of KMU-1170 as an anti-inflammatory agent, we analyzed the potential superiority of the anti-inflammatory efficacy of KMU-1170 by comparing it with various other anti-inflammatory drugs. At 1 μM, KMU-1170 significantly suppressed LPS-induced increase of IL-1β protein in THP-1 cells, whereas this effect was not observed even at higher concentrations of other tested drugs, such as celecoxib (COX-2 inhibitor), ibrutinib (BTK inhibitor), ruxolitinib (JAK inhibitor) and methotrexate (Anti-inflammatory drug of RA) (Figure 5C,D). Only 1 μM methotrexate inhibited LPS-induced upregulation of IL-1β to a similar degree as KMU-1170 did (Figure 5C,D).

### 2.8. KMU-1170 Suppresses LPS-Induced Inflammation in Osteoarthritic FLS

Lastly, we investigated whether KMU-1170 inhibits inflammatory signal transduction in osteoarthritis (OA) using human osteoarthritic FLS. We obtained human osteoarthritic FLS from the synovial tissues of patients with knee OA. Pretreatment of FLS with KMU-1170 inhibited LPS-induced upregulation of IL-1β, TNF-α, and IL-6 mRNA and protein levels (Figure 6A–D) and iNOS and COX-2 protein levels (Figure 6E). Furthermore, the KMU-1170 pretreatment inhibited LPS-induced phosphorylation of IKKα/β and NF-κB p65 in FLS (Figure 6F).

## 3. Discussion

Protein kinases are enzymes that phosphorylates protein molecules by attaching a phosphoryl group to serine, threonine, or tyrosine residue, and binding with ATP is essential for kinase activity [18]. Accumulating evidence has recently indicated that protein kinases are implicated in various human disease including cancer and inflammatory diseases [12,19], resulting in development of the FDA-approved protein kinase inhibitors [20]. In this study, we discovered a novel multi-target protein kinase inhibitor KMU-1170 and identified its anti-inflammatory mechanisms in both THP-1 cells and primary human osteoarthritic FLS.

KMU-1170, indolin-2-one derivative, was found during the screening of the in-house protein kinase inhibitor library for anti-inflammatory activities. Indolin-2-one is regarded as a privileged scaffold for the discovery of kinase inhibitors [21]. Antitumor drugs such as sunitinib and nintedanib are representative kinase inhibitors that were developed using the indolin-2-one scaffold [21]. A series of indolin-2-one analogues were synthesized based on a de novo design for the inhibitor of mitogen activated protein kinase 1 (MAPK1). Indolin-2-one was designed to bind at the ATP binding pocket of MAPK1 using two H-bond interactions. Two substituents were attached at the 3- and 5-postion of indolin-2-one to increase binding affinity: pyrazine at the 5-positon for the interaction with the ε-amino group of Lys54, imidazole at the 3-position for the H-bond interaction with the backbone carbonyl group of Met108 of MAPK1 (PDB code: 3W55). The introduction of cyclopropylamine at the 5-position of pyrazine yielded KMU-1170. The novel compound was found to strongly inhibit the activities of inflammation-associated protein kinases (Table 1).

A critical mediator of inflammatory responses is iNOS, which belongs to a family of catalytic enzymes is necessary for nitric oxide (NO) production [22]. Because excessive NO produced by iNOS is involved in various pathophysiological conditions [23], iNOS activity/level must be controlled. We found that KMU-1170 inhibited LPS-induced upregulation of iNOS in THP-1 cells (Figure 1C,D). COX, which is composed of two isozymes, COX-1 and COX-2, is associated with diverse human diseases, such as inflammatory diseases, neoplastic diseases, and Alzheimer’s disease [24]. Selective inhibition of COX-2 represents a breakthrough therapeutic strategy that has been found to control inflammation and pain while producing few side effects such as gastrointestinal irritation [24]. Therefore, we investigated the inhibitory effect on COX-2 in both THP-1 cells and human osteoarthritic FLS, and found that KMU-1170 selectively suppressed LPS-induced upregulation of COX-2 in these cells (Figure 1F,G and Figure 6E).

TAK1 is associated with various cellular processes, including cell death, differentiation, immunity, inflammation, and survival [25,26,27]. Notably, TAK1 acts upstream of NF-κB and MAPKs in LPS-TLR4 signaling-mediated inflammation process [28,29]. Additionally, phosphorylation of TAK1 at Ser412 is critical for LPS-TLR4-TAK1 signaling-mediated production of proinflammatory cytokines [30]. MAPK pathways are widely recognized to play critical roles in TLR-mediated inflammatory responses, such as the production of proinflammatory cytokines stimulated by LPS [31]. Here, we found that KMU-1170 suppressed LPS-induced phosphorylation of TAK1, ERK, and JNK in THP-1 cells (Figure 3C–F).

The transcription factor NF-κB is a major mediator in the immune system that regulates key factors, such as cytokines and the inflammasome, in an inflammatory response [31]. NF-κB protein is sequestered in the cytoplasm through its binding to inhibitor of NF-κB (IκB) [31]. Moreover, in the canonical NF-κB pathway, phosphorylation of IκB by IKK results in IκB degradation through ubiquitination, following which the NF-κB complex enters the nucleus and acts as a transcription factor to enable the expression of specific genes involved in the inflammatory response [31]. Notably, upon LPS binding, the TLR-MyD88 signal activates the transcription factor NF-κB, which then leads to the induction of proinflammatory cytokines such as IL-1β, TNF-α, and IL-6 [32,33]. Our results showed that KMU-1170 exerts inhibitory effects on LPS-induced phosphorylation of IKKα/β and NF-κB p65, nuclear translocation of NF-κB p65, and upregulation of IL-1β, TNF-α, and IL-6 (Figure 2, Figure 3C,D and Figure 4). However, KMU-1170 did not inhibit LPS-mediated upregulation of TLR4 and MyD88 in THP-1 cells (Figure 3A). Therefore, these results suggest that KMU-1170 inhibits inflammatory signal transduction downstream of TLR4/MyD88 in the LPS-mediated inflammatory response of THP-1 cells.

The inflammasome is a multiprotein complex that is considered to function in the recognition of pathogen-associated molecular patterns and danger-associated molecular patterns in the inflammatory responses of the innate immune system [34]. In the two-signal model, the NLRP3 inflammasome, which consists of NLRP3, ASC, and pro-caspase-1, is regulated by a priming signal and an activation signal [10]. The first signal, the priming signal, is induced by endogenous cytokines or microbial molecules such as LPS, and this leads to the upregulation of NLRP3 and pro-IL-1β through NF-κB activation. Next, the second signal, the activation signal, is initiated by a plethora of stimuli, such as ATP, and activates the NLRP3 inflammasome [10]. In this study, we investigated whether KMU-1170 can inhibit the two signals in THP-1 cells, which revealed that KMU-1170 attenuated LPS-induced upregulation of NLRP3 and pro-IL-1β as well as ATP-mediated production of IL-1β (Figure 5A,B).

Recently, various signaling pathways have been identified as therapeutic targets for inflammation, and numerous anti-inflammatory agents are currently under development [35]. Here, we sought to compare the anti-inflammatory efficacy of KMU-1170 with that of other targeted anti-inflammatory drugs. KMU-1170 exhibited superior inhibitory efficiency in inflammatory signaling pathway relative to other drugs, including a COX-2 inhibitor (celecoxib), a BTK inhibitor (ibrutinib), a JAK inhibitor (ruxolitinib), and methotrexate (Figure 5C,D); however, because KMU-1170 exerts a multi-kinase inhibitory effect (Table 1), additional studies will be required before the application of this inhibitor to a multicellular system.

Lastly, we investigated whether KMU-1170 suppresses LPS-mediated inflammatory responses in human osteoarthritic FLS. Osteoarthritic FLS play a pivotal role in the processes of OA characterized by continuous synovial inflammation and progressive cartilage degradation [36]. Moreover, macrophage-mediated inflammation play an important role in pathophysiology of OA [37,38]. As shown in Figure 6, KMU-1170 inhibited LPS-induced inflammatory responses in human osteoarthritic FLS, suggesting that KMU-1170 can alleviate OA exacerbation.

In order to develop the KMU-1170 as a promising anti-inflammatory agent, it is necessary to improve therapeutic window of concentration, reduce toxicity, improve the efficacy on various cell model of inflammatory disease, and conduct in vivo study. Nevertheless, we expected that KMU-1170, which has the inhibitory effect on LPS-mediated inflammatory pathway, will be a candidate agent for the treatment of inflammatory diseases.

In conclusion, this study demonstrated that a novel multi-protein kinase inhibitor, KMU-1170, suppresses inflammatory signal transduction by controlling the p-TAK1/p-IKKα/β/p-NF-κB/proinflammatory cytokine signaling pathway and the NLRP3 inflammasome (Figure 7). Our findings suggest that KMU-1170 is a highly favorable compound for development as a potent anti-inflammatory drug.

## 4. Materials and Methods

### 4.1. Synthesis of KMU-1170

#### 4.1.1. General Information

All reactions were performed using commercially available reagents and solvents without further purification, under proper conditions as stated by the manufacturers. A CEM Discover BenchMate was used for microwave-assisted reactions. Reaction completion was monitored on E. Merck silica gel F254 TLC plates. Synthesized compounds were purified by means of flash column chromatography using Merck Silica Gel 60 (230–400 mesh). Synthesized compounds were characterized by performing ^1^H-NMR on a JEOL ECA 500 (1H: 500 MHz) spectrometer and chemical shift δ values were measured in ppm with TMS as the reference standard. Mass spectra were obtained using a Waters ACQUITY UPLC, Micromass Quattro micro^TM^ API.

#### 4.1.2. 5-(4,4,5,5-Tetramethyl-1,3,2-dioxaborolan-2-yl)indolin-2-one (**1**)

A microwave vessel was filled with 5-bromoindoline-2,3-dione (0.50 g, 2.2 mmol), hydrazine hydrate (0.14 g, 4.4 mmol), and ethanol (EtOH, 2 mL). The mixture was irradiated for 10 min at 100 °C by applying 100 W. After addition of sodium hydroxide (0.18 g, 4.4 mmol), the mixture was irradiated for 10 min at 80 °C by applying 100 W. The mixture was acidified using 2 M hydrochloric acid (HCl), and the precipitate formed by the addition of cold water was collected through filtration and washed with 2 M HCl and then with water. Drying under vacuum generated 0.42 g of 5-bromoindolin-2-one (at 89% yield), which was used for the next step without further purification. A microwave vessel was filled with 5-bromoindolin-2-one (0.40 g, 1.9 mmol) and 1,4-dioxane (2.0 mL), and after adding bis(pinacolato)diboron (0.72 g, 3.8 mmol) and potassium acetate (KOAc; 0.56 g, 5.7 mmol) to the mixture, the mixture was purged with N_2_ gas for 5 min. To the reaction mixture was added 1,1′-bis(diphenylphosphino)ferrocene]dichloropalladium(II) (PdCl_2_(dppf)) (0.069 g, 0.094 mmol) in N_2_ gas, and the mixture was irradiated for 10 min at 110 °C by applying 100 W. After solvent removal in vacuo, the residue was purified by means of silica gel chromatography using 1:1 hexane (HEX)/ethyl acetate (EA), which yielded 0.41 g (82%) of the title compound.

#### 4.1.3. 6-Chloro-N-cyclopropylpyrazin-2-amine (**2**)

Cyclopropylamine (5.1 mmol, 0.35 mL) and potassium carbonate (K_2_CO_3_; 6.7 mmol, 0.93 g) were added to 5.0 mL of *N*,*N*-dimethylformamide, and the mixture was stirred for 30 min at room temperature. After addition of 2,6-dichloropyrazine (3.4 mmol, 0.50 g), the reaction mixture was stirred overnight at room temperature, and then the solvent was removed in vacuo. The residue was next treated with dichloromethane (DCM) and filtered with the aid of celite, and the filtrate was collected and the solvent was removed in vacuo. The final residue was purified through silica gel chromatography by using 5:1 DCM/EA, which yielded 0.21 g (36%) of the title compound.

#### 4.1.4. 5-(6-(Cyclopropylamino)pyrazin-2-yl)indolin-2-one (**3**)

A microwave vessel was filled with compound **1** (0.45 g, 1.8 mmol), compound **2** (0.20 g, 1.2 mmol), 1,4-dioxane (2.0 mL), EtOH (0.40 mL), and 2 M aqueous potassium carbonate (1.8 mL, 2.54 mmol). After adding tetrakis (triphenylphosphine) palladium(0) (PdCl_2_(PPh_3_)_2_) (0.041 g, 0.059 mmol) in an N_2_ atmosphere, the mixture was irradiated for 10 min at 110 °C by applying 100 W. The solvent was removed in vacuo, and the residue was treated with DCM and then filtered with the aid of celite. The filtrate was collected and the solvent was removed in vacuo. The residue was purified by performing silica gel chromatography with 5:1 DCM/EA, which yielded 0.19 g (41%) of the title compound.

#### 4.1.5. (Z)-3-((1H-Imidazol-5-yl)methylene)-5-(6-(cyclopropylamino)pyrazin-2-yl)indolin-2-one (KMU-1170)

A microwave vessel was filled with compound 3 (0.15 g, 0.056 mmol), piperidine (6.0 μL, 0.06 mmol), 1*H*-imidazole-4-carbaldehyde (0.081 g, 0.85 mmol), and EtOH (2.0 mL), and the mixture was irradiated for 10 min at 80 °C by applying 100 W. The solvent was removed in vacuo and the residue was purified by performing silica gel chromatography with 5:95 methanol/DCM, which yielded 0.042 g (22%) of the title compound. ^1^H-NMR (500 MHz, DMSO-d_6_) δ values: 11.27–11.14 (br, 1H), 8.41 (s, 1H), 8.38 (s, 1H), 8.06 (s, 1H), 7.99 (s, 1H), 7.96 (d, *J* = 8.0 Hz, 1H), 7.95 (s, 1H), 7.69 (s, 1H), 7.29 (s, 1H), 7.01 (d, *J* = 8.0 Hz, 1H), 2.71 (s, 1H), 0.79 (m, 2H), 0.57–0.45 (m, 2H). ESI MS: m/z = 345 [M + H]^+^.

### 4.2. Reagents

LPS (*Escherichia coli* serotype) was purchased from Sigma-Aldrich (St. Louis, MO, USA). Celecoxib, ibrutinib, methotrexate, and ruxolitinib were purchased from Selleck Chemicals (Houston, TX, USA). Primary antibodies against iNOS, TLR4, MyD88, p-TAK1 (Ser412), p-ERK, ERK, p-JNK, p-IKKα/β, IKKα/β, p-NF-κB p65, NF-κB, ASC, caspase-1, and NLRP3 were purchased from Cell Signaling Technology (Beverly, MA, USA). Anti-IL-1β antibody was purchased from Novus Biologicals (Centennial, CO, USA). Antibodies against IL-6, COX-1, COX-2, TAK1, p-p38, and p38 were purchased from Santa Cruz Biotechnology (Santa Cruz, CA, USA) and Sigma-Aldrich (St. Louis, MO, USA), respectively. Anti-TNF-α antibody was purchased from Abcam (Cambridge, MA, USA). Anti-β-actin antibody was purchased from Sigma-Aldrich (St. Louis, MO, USA). Anti-mouse IgG-horseradish peroxidase (HRP) and anti-mouse IgG-HRP antibodies were purchased from Santa Cruz Biotechnology (Santa Cruz, CA, USA).

### 4.3. Cell Line and Culture

The human monocytic cell line THP-1 cells (Korean Cell Line Bank, Seoul, Korea) were cultured in RPMI1640 medium (Welgene Inc., Gyeongsan, Korea), 10% fetal bovine serum (FBS; Welgene Inc., Gyeongsan, Korea), and 1% antibiotic-antimycotic solution (Welgene Inc., Gyeongsan, Korea). The cells were cultured at 37.5 °C, 5% CO_2_ in fully humidified air.

### 4.4. Isolation and Culture of Primary Human Osteoarthritic FLS

Isolation of human osteoarthritic FLS was performed by enzymatic digestion of synovial tissues from osteoarthritic patients who underwent a synovectomy at Keimyung University Dongsan Medical Center. The enrolled patients with OA fulfilled the American College of Rheumatology criteria and agreed to give their informed consent. The experimental study was approved by the Institutional Review Board (IRB) of Keimyung University Dongsan Medical Center on November 20th, 2020 [IRB No. 2020-11-031]). After crushing the osteoarthritic tissues into 2–3-mm pieces, the minced tissues were treated with 0.5 mg/mL type II collagenases (Thermo Fisher Scientific, Waltham, MA, USA) in Dulbecco’s modified Eagle medium (DMEM; Welgene Inc., Gyeongsan, Korea) under 5% CO_2_ at 37.5 °C for 2 h. The separated cells were centrifuged (3000 rpm, 5 min) and then resuspended in DMEM), 10% fetal bovine serum (FBS; Welgene Inc., Gyeongsan, Korea), and 1% antibiotic-antimycotic solution (Welgene Inc., Gyeongsan, Korea), and agitated in a 75-cm^2^ flask. One day later, the adherent cells were incubated under 5% CO_2_ at 37.5 °C in DMEM), 10% fetal bovine serum (FBS; Welgene Inc., Gyeongsan, Korea), and 1% antibiotic-antimycotic solution (Welgene Inc., Gyeongsan, Korea), and the culture fluid was replaced every 3 days. When 90%–95% of the bottom of the flask was filled with fibroblasts, the medium was diluted 1:3 with fresh culture.

### 4.5. Kinase-Profiling Analysis

To test the specificity of a novel multi-protein kinase inhibitor KMU-1170, a kinase-profiling service was conducted by Eurofins Cerep S.A. (Celle-Lévescault, France). The kinase-profiling analysis was performed with 1 μM of compound and a concentration of ATP at the Km value for each individual kinase and kinase substrate according to Eurofin’s protocols.

### 4.6. Cell Viability Assay

The cell viability analysis was performed using XTT assay (Welgene Inc., Gyeongsan, Korea). Briefly, 2 × 10^5^ THP-1 cells were plated in 96-well plates for 24 h. Cells were treated with various concentrations of KMU-1170 and incubated under 5% CO_2_ at 37 ℃ for 24 h. Subsequently, 0.5 mg/mL XTT solution was added to the culture solution and then incubated under 5% CO_2_ at 37 ℃ for 3 h. Measurement of optical density (OD) was performed using a microplate reader (BMG Labtech, Ortenberg, Germany) at a 450 nM wavelength.

### 4.7. Western Blotting Analysis

THP-1 cells (2 × 10^6^ cells/well in 60-mm dishes) and FLS (2 × 10^6^ cells/well in 100-mm dishes) were plated for western blotting analysis. After culturing for 24 h, cells were lysed in RIPA buffer (20 mM HEPES and 0.5% Triton X-100, pH 7.6) and supernatant fractions were collected. Protein concentrations were measured using a BCA assay kit (Thermo Fisher Scientific, Waltham, MA, USA). Equal amounts of protein were electrophoresed in SDS-PAGE and transferred to nitrocellulose membranes (GE Healthcare Life Science, Pittsburgh, PA, USA). Subsequently, the membranes were incubated with blocking buffer (0.05% Tween 20 with 5% non-fat dry milk in Tris-buffered saline (TBS)) for 1 h and then incubated for 24 h with appropriately diluted primary antibodies against specific target proteins. Next, the membranes were washed with TBS with Tween 20, and then incubated at room temperature for 1 h with the appropriate secondary antibodies. The specific proteins were detected with ECL western blotting kit (EMD Millipore, Darmstadt, Germany) according to the manufacturer’s protocols, and the signal strength was measured using a Chemi Image System Fusion FX (Vilber Lourmat, Collégien, France). Protein bands were quantified using Image-J software. Quantitative analysis of band density using the Image-J is shown as the mean ± standard deviation (SD) of three independent experiments.

### 4.8. RNA Isolation and Reverse Transcription-Polymerase Chain Reaction (RT-PCR) and Real Time Quantitative PCR (qPCR)

Total cellular RNA was isolated using TRIzol Reagent (Thermo Fisher Scientific, Waltham, MA, USA). Each RNA was quantified using a NanoDrop 1000 (Thermo Fisher Scientific, Waltham, MA, USA). Each cDNA was obtained from 1 μg of the isolated RNA using Moloney murine leukemia virus reverse transcriptase (Gibco-BRL, Gaithersburg, MD, USA) according to the manufacturer’s instructions. The primer sequences used in the study are shown in Table 2. For PCR, DNA polymerase was used with primers targeting IL-1β, TNF-α, and IL-6. The amplified PCR products were separated by electrophoresis on a 2% agarose gel and detected under ultraviolet light. For qPCR, by using the specific primer and SYBR GREEN Premix (Toyobo, Osaka, Japan), qPCR was performed on the LightCycler^®^ 480 real-time PCR system (Roche Diagnostics, Mannheim, Germany). The experiments were performed three times, and the threshold cycle number (Ct) of each target mRNA was normalized to β-actin. The delta-delta Ct values of each genes are indicated as the mean ± SD of three independent experiments.

### 4.9. Immunofluorescence Staining

THP-1 cells (1 × 10^3^) were cultured in 8-chamber glass slides for 24 h, treated for 1 h with KMU-1170 at 1 μmol/L, and then stimulated with LPS for 6 h. The cells were washed with phosphate-buffered saline (PBS), fixed with 4% formaldehyde, permeabilized using 0.2% Triton X-100 in PBS, and incubated for 1 h with bovine serum albumin to block nonspecific binding. Subsequently, the cells were incubated with primary antibodies at 4 °C for 24 h and then with FITC-conjugated goat anti-mouse IgG (Thermo Fisher Scientific, Waltham, MA, USA) and 4′,6-diamidino-2-phenylindole (Thermo Fisher Scientific, Waltham, MA, USA; for nuclear staining), after which the stained cells were examined under a fluorescence microscope.

### 4.10. Statistical Analysis

Quantitative results are presented as the mean and SDs. The data were analyzed by a one-way ANOVA and post-hoc comparisons (Student-Newman-Keuls) using the Statistical Package for Social Science 26.0 software (SPSS Inc.; Chicago, IL, USA). A *p* value less than or equal to 0.05 was considered significant.

## Figures and Tables

**Figure 1 ijms-22-01194-f001:**
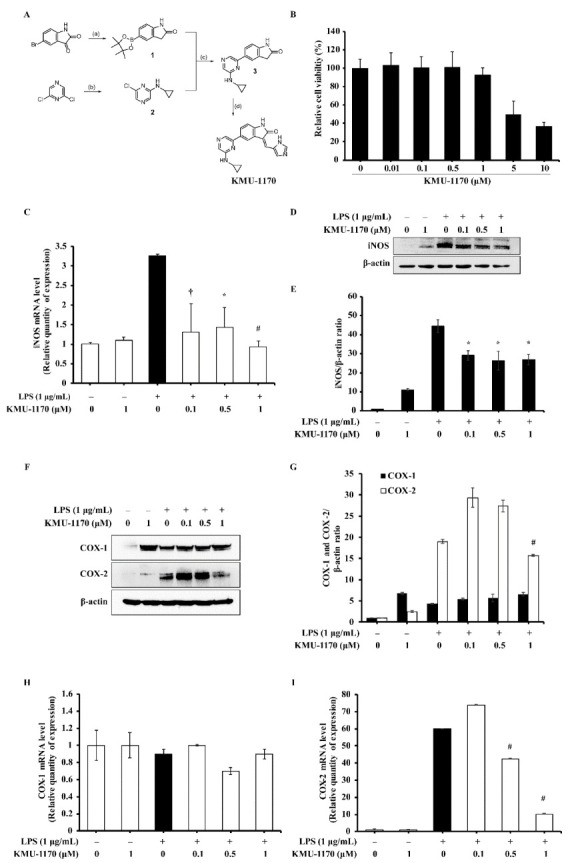
Discovery of KMU-1170, and the effect of KMU-1170 on LPS-induced iNOS and COX-2 in THP-1 cells. (**A**) Synthetic scheme for the synthesis of KMU-1170. Reagents and experimental conditions: (a) (i) NH_2_NH_2_, NaOH, EtOH; (ii) bis (pincholato)diboron, PdCl_2_(dppf), KOAc, 1,4-dioxane: EtOH(1:1.5), microwave; (b) K_2_CO_3_, cyclopropylamine, DMF; (c) PdCl_2_(PPh_3_)_2_, 2M K_2_CO_3_, 1,4-dioxane: EtOH (1:1.5), microwave; (d) piperidine, 1*H*-imidazole-4-carbaldehyde, EtOH, microwave. (**B**) Cytotoxic effect of KMU-1170 in THP-1 cells was performed by XTT assay. Cell were differentiated into macrophages for 24 h using PMA (100 nM), then the cells were treated with different doses of KMU-1170 (0.01, 0.1, 0.5, 1, 5, and 10 μM) for 24 h. (**C**–**I**) Cells were treated with LPS (1 μg/mL) for 6 h after pretreatment with different doses of KMU-470 (0.1, 0.5, and 1 μM) for 1 h. Total RNA was extracted, and used to determine the mRNA expression levels of iNOS, COX-1, and COX-2 (**C**,**H**,**I**, respectively). Whole cell lysates were isolated and used to measure the protein expression levels of iNOS and β-actin by Western blot analysis (**D**). Image-J software was used to analyze the relative optical density of the iNOS band (**E**). Whole cell lysates were isolated and used to measure the protein expression levels of COX-1, COX-2, and β-actin by Western blot analysis (**F**). Image-J software was used to analyze the relative optical density of the COX-1 and Cox-2 band (**G**). † *p* < 0.05, * *p* < 0.01, and # *p* < 0.001 compared to LPS alone.

**Figure 2 ijms-22-01194-f002:**
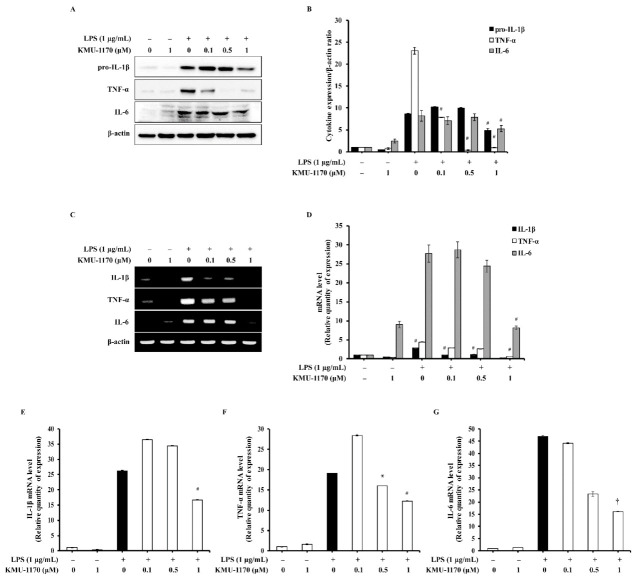
Inhibitory effect of KMU-1170 on LPS-induced upregulation of proinflammatory cytokines in THP-1 cells. Cells were differentiated into macrophages for 24 h using PMA (100 nM). And then the cells were treated with LPS (1 μg/mL) for 6 h after pretreatment with different doses of KMU-1170 (0.1, 0.5, and 1 μM) for 1 h. (**A**) Whole cell lysates were isolated and used to measure the protein expression levels of pro-IL-1β, TNF-α, IL-6, and β-actin by Western blot analysis. (**B**) Image-J software was used to analyze the relative optical density of the pro-IL-1β, TNF-α, and IL-6 band, respectively. (**C**) Total RNA was extracted and used to determine the mRNA expression levels of IL-1β, TNF-α, and IL-6 using RT-PCR. (**D**) Image-J software was used to analyze the relative optical density of the IL-1β, TNF-α, and IL-6 band, respectively. (**E**–**G**) Total RNA was extracted and used to determine the mRNA expression levels of IL-1β, TNF-α, and IL-6 using real time PCR. † *p* < 0.05, * *p* < 0.01, and # *p* < 0.001 compared to LPS alone.

**Figure 3 ijms-22-01194-f003:**
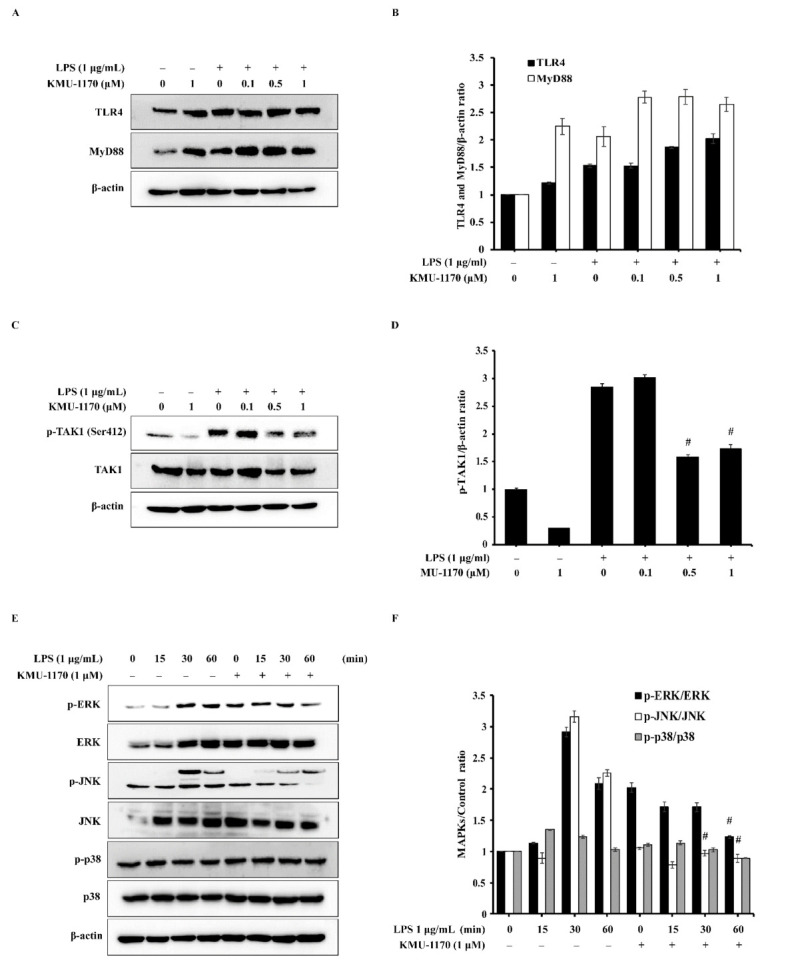
Effect of KMU-1170 on LPS-TLR4 signaling pathway-related proteins in THP-1 cells. (**A**,**C**) Cells were differentiated into macrophages for 24 h using PMA (100 nM). And then the cells were treated with LPS (1 μg/mL) for 6 h after pretreatment with different doses of KMU-1170 (0.1, 0.5, and 1 μM) for 1 h. Whole cell lysates were isolated and used to measure the protein expression levels of TLR, MyD88, and β-actin by Western blot analysis (**A**). (**B**) Image-J software was used to analyze the relative optical density of the TLR4 and MyD88 band. Whole cell lysates were isolated and used to measure the protein expression levels of p-TAK1, TAK1, and β-actin by Western blot analysis (**C**). (**D**) Image-J software was used to analyze the relative optical density of the p-TAK1 band. (**E**) Cells were differentiated into macrophages for 24 h using PMA (100 nM), then the cells were treated with or without 1 μM KMU-1170 for 24 h and stimulated with LPS (1 μg/mL) at the indicated times. Whole cell lysates were isolated and used to measure the protein expression levels of p-ERK, ERK, p-JNK, JNK, p-p38, p38, and β-actin by Western blot analysis. (**F**) Image-J software was used to analyze the relative optical density of the p-ERK, p-JNK, and p-p38 band. # *p* < 0.001 compared to LPS alone.

**Figure 4 ijms-22-01194-f004:**
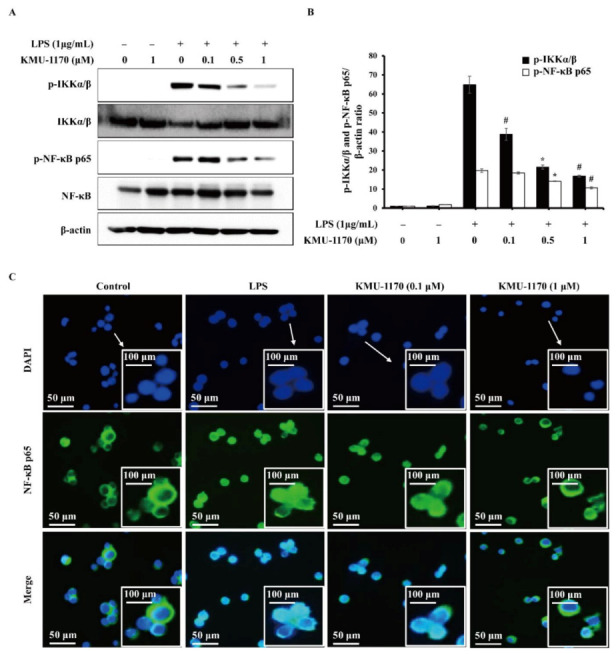
Inhibitory effect of KMU-1170 on LPS-induced phosphorylation of IKKα/β and NF-κB, and nuclear translocation of NF-κB in THP-1 cells. Cells were differentiated into macrophages for 24 h using PMA (100 nM). And then the cells were treated with LPS (1 μg/mL) for 6 h after pretreatment with the indicated concentrations of KMU-1170 for 1 h. (**A**) Whole cell lysates were isolated and used to measure the protein expression levels of p-IKKα/β, IKKα/β, p-NF-κB p65, NF-κB p65, and β-actin by Western blot analysis. (**B**) Image-J software was used to analyze the relative optical density of the p-IKKα/β and p-NF-κB p65 band. (**C**) Cells were stained with antibodies to the NF-κB p65 (**green**) and 4′,6-diamidino-2-phenylindole (**blue**) and captured at ×200 using fluorescence microscope. Arrows indicate 400 times magnification. * *p* < 0.01 and # *p* < 0.001 compared to LPS alone.

**Figure 5 ijms-22-01194-f005:**
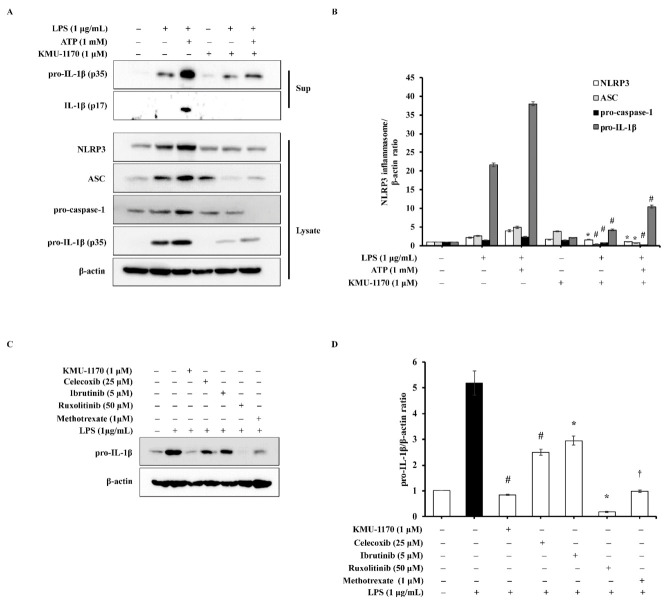
Inhibitory effect of KMU-1170 on LPS-induced activation of NLRP3 inflammasome and anti-inflammatory potentiality of KMU-1170 in THP-1 cells. (**A**) Cells were differentiated into macrophages for 24 h using PMA (100 nM). And then the cells were treated with LPS (1 μg/mL) and/or ATP (1 mM) for 6 h after pretreatment with 1 μM KMU-1170 for 1 h. Cell lysate (Lysate) and media supernatant (Sup) were isolated and used to measure the protein expression levels of pro-IL-1β and IL-1β for the Sup as well as NLRP3, ASC, pro-caspase-1, pro-IL-1β, and β-actin for the Lysate by Western blot analysis. (**B**) Image-J software was used to analyze the relative optical density of the NLRP3, ASC, pro-caspase-1, and pro-IL-1β band in the Lysate. (**C**) Cells were differentiated into macrophages for 24 h using PMA (100 nM). And then the cells were treated with LPS (1 μg/mL) for 6 h after pretreatment with KMU-1170 (1 μM), celecoxib (25 μM), ibrutinib (5 μM), luxolitinib (50 μM), and metotrexate (1 μM) for 1 h. Whole cell lysates were isolated and used to measure the protein expression levels of pro-IL-1β and β-actin by Western blot analysis. (**D**) Image-J software was used to analyze the relative optical density of the pro-IL-1β band. † *p* < 0.05, * *p* < 0.01, and # *p* < 0.001 compared to LPS alone.

**Figure 6 ijms-22-01194-f006:**
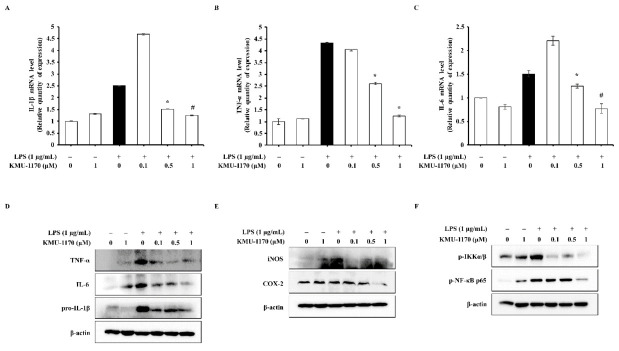
Inhibitory effect of KMU-1170 on LPS-mediated inflammatory response in primary human osteoarthritic FLS. Cells were treated with LPS (1 μg/mL) for 6 h after pretreatment with different doses of KMU-1170 (0.1, 0.5, and 1 μM) for 1 h. (**A**–**C**) Total RNA was extracted and used to determine the mRNA expression levels of IL-1 β, TNF-α, and IL-6 using real time PCR. (**D**) Whole cell lysates were isolated and used to measure the protein expression levels of pro-IL-1β, TNF-α, IL-6, and β-actin by Western blot analysis. (**E**) Whole cell lysates were isolated and used to measure the protein expression levels of iNOS, COX-2, and β-actin by Western blot analysis. (**F**) Whole cell lysates were isolated and used to measure the protein expression levels of p-IKKα/β, p-NF-κB p65, and β-actin by Western blot analysis. * *p* < 0.01 and # *p* < 0.001 compared to LPS alone.

**Figure 7 ijms-22-01194-f007:**
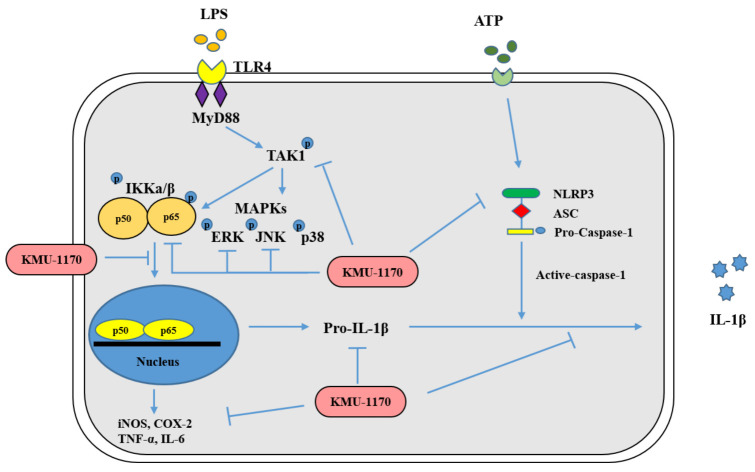
Mechanistic scheme summarizing the anti-inflammatory effect of KMU-1170. LPS, lipopolysaccharide; TLR4, toll-like receptor 4; MyD88, myeloid differentiation factor 88; TAK1, transforming growth factor-β-activated kinase 1; IKK, inhibitor NF-κB kinase; MAPKs, mitogen-activated protein kinases; NF-κB, nuclear factor-κB; iNOS, inducible nitric oxide synthase; COX-2, cyclooxygenase-2; TNF-α, tumor necrosis factor-α; IL-1β, interleukin-1β; NLRP3, NOD-like receptor family, pyrin domain containing 3; ASC, apoptosis-associated speck-like protein containing a CARD.

**Table 1 ijms-22-01194-t001:** Kinase activity of 15 protein kinases upon treatment with 1 μM KMU-1170.

KMU-1170 1 μM	
Kinase	Activity in % *
MAPK1 (h)	2
Yes (h)	23
Blk (h)	20
Fgr (h)	29
Lyn (h)	13
Lck (h)	7
TYK2 (h)	6
JAK3 (h)	3
Fyn (h)	26
Itk (h)	45
Syk (h)	55
JAK2 (h)	41
JNK1α1 (h)	67
Hck (h)	30
Pyk2 (h)	67
cSRC (h)	36
Bmx (h)	26
Txk (h)	−7
Tec (h) activated	41
SAPK2a (h)	108
BTK (h)	40
ZAP-70 (h)	108

* Values were obtained from the KinaseProfilerTM project of Eurofins.

**Table 2 ijms-22-01194-t002:** Primer sequences for PCR and Real Time PCR.

Primers	Sequences (5′ → 3′)	
iNOS	Forward	CTG TCT GGT TCC TAC GTC ACC
	Reverse	CCC ACG TTA CAT GGG AGG ATA
COX-1	Forward	ACC TTG AAG GAG TCA GGC ATG AG
	Reverse	TGT TCG GTG TCC AGT TCC AAT A
COX-2	Forward	ATC ACA GGC TTC CAT TGA CC
	Reverse	TAT CAT CTA GTC CGG AGG GG
IL-1β	Forward	CCT TGG GCC TCA AGG AAA A
	Reverse	CTC CAG CTG TAG AGT GGG CTT A
TNF-α	Forward	GGA GAA GGG TGA CCG ACT CA
	Reverse	CTG CCC AGA CTC GGC AA
IL-6	Forward	ATG GCA CAG TAT CTG GAG GAG
	Reverse	TAA GCT GGA CTC ACT CTC GGA
β-actin	Forward	AAT CTG GCA CCA CAC CTT CTA
	Reverse	ATA GCA CAG CCT GGA TAG CAA

## Data Availability

The data used in the current study are available from the corresponding authors upon reasonable request.

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
