# Peer review of "KMU-1170, a Novel Multi-Protein Kinase Inhibitor, Suppresses Inflammatory Signal Transduction in THP-1 Cells and Human Osteoarthritic Fibroblast-Like Synoviocytes by Suppressing Activation of NF-κB and NLRP3 Inflammasome Signaling Pathway"

_ijms, 2021, doi:10.3390/ijms22031194_

Round 1
Reviewer 1 Report
In this work, the authors study the anti-inflammatory properties of a novel multi-protein kinase inhibitor KLMU-1170. The study seems to be well done. The English is acceptable but the text would gain from some corrections. The statistical analysis needs complete revision.
1) Very often it is not clear what the authors are comparing. For instance, at the end of the legend to figure 1, the state “…compared to LPS”. Which LPS do they mean ? LPS with 0 uM KMU-1170 ? For instance in Fig. 1G, there is only one significant comparison and it is impossible to know to what it is compared. Are 0.1 and 0.5 uM not different from 0 uM for COX 2 ?
Fig 2F. Why is TNFa mRNA increased at 0.1 uM KMU-1170 ? This looks statistically different from 0 KMU-1170 but it is considered non significant.
I could make such remarks for nearly every figure .
Some data require multivariate analysis (Figure 2 B and D for instance and many others)
2) At the beginning of the “Discussion” (lines 238…) they justify the use of the indolin scaffold. They should also justify the rational for the modifications they make on this scaffold.
3) Figure 7: Inhibitory line going from KMU-1170 to the nucleus. It is not clear what is the target for inhibition
Minor points:
Line 15: which are
Line 58: essential
Line 230: Protein kinases are enzymes that
Line 239: indolin-2-one
Line 474: one-way ANOVA
Author Response
#1 reviewer :
In this work, the authors study the anti-inflammatory properties of a novel multi-protein kinase inhibitor KLMU-1170. The study seems to be well done. The English is acceptable but the text would gain from some corrections. The statistical analysis needs complete revision.
1) Very often it is not clear what the authors are comparing. For instance, at the end of the legend to figure 1, the state “…compared to LPS”. Which LPS do they mean ? LPS with 0 uM KMU-1170 ? For instance in Fig. 1G, there is only one significant comparison and it is impossible to know to what it is compared. Are 0.1 and 0.5 uM not different from 0 uM for COX 2 ?
Answer: Thank you for your kind advice. As you mentioned, we carefully checked the results. ‘LPS’ meant ‘LPS treatment alone’. Therefore, we modified ‘.compared to LPS’ into ‘compared to LPS alone’ in all statistical illustrations.
Fig 2F. Why is TNFa mRNA increased at 0.1 uM KMU-1170 ? This looks statistically different from 0 KMU-1170 but it is considered non significant.
I could make such remarks for nearly every figure .
Answer: Thank you for your kind advice. As you mentioned, we carefully checked the results. Actually, we conducted several experiments and found that KMU-1170 (0.1μM) increased mRNA expression compared to LPS alone. To identify the reason, we carefully explored the other papers investigating this phenomenon. We found another paper showing that low concentrations of drugs increase the inflammatory cytokine mRNA expression levels compared to LPS alone (1). In the study, researchers did not also discuss the phenomenon.
As shown in Table 1, KMU-1170 exhibits inhibitory effect on multi-protein kinases, but at low concentration such as 0.1 mM, it is not known which kinase is the main target. Therefore, through additional research, we will investigate why low concentration of compounds increase the expression of specific cytokines. Although not all data in this study showed a concentration-dependent pattern, we would like to place great significance that KMU-1170 (1 mM) has anti-inflammatory effect in LPS-mediated inflammatory responses in THP-1 cells and osteoarthritic fibroblast-like synoviocytes.
- Guo F, Ding Y, Yu X, Cai X. Effect of dexmedetomidine, midazolam, and propofol on lipopolysaccharide-stimulated dendritic cells. Exp Ther Med. 2018;15(6):5487-94.
Some data require multivariate analysis (Figure 2 B and D for instance and many others)
Answer: Thank you for your kind advice. As you mentioned, we carefully checked the results. Actually, Figure 2B is analyzing results of Image-J software and indicated that the analyzed results of relative optical density of the pro-IL-1β, TNF-α, and IL-6 band of Figure 2A. Additionally, Figure 2D is also analyzing results of Image-J software and indicated that the analyzed results of relative optical density of the pro-IL-1β, TNF-α, and IL-6 band of Figure 2C. In order to efficiently represent the analysis results, the analysis results of each cytokine was derived and then we plotted the results into a single graph. To clarify this meaning, we added ‘respectively’ in the revised manuscript. You can see the modified descriptions in the revised manuscript (page 6, line 128 and 130).
2) At the beginning of the “Discussion” (lines 238…) they justify the use of the indolin scaffold. They should also justify the rational for the modifications they make on this scaffold.
Answer: Thank you for your kind advice. As you mentioned, we added justification of use of the indolin scaffold and the rationale for the modifications. You can see the modified descriptions in the revised manuscript (page 11, line 242).
3) Figure 7: Inhibitory line going from KMU-1170 to the nucleus. It is not clear what is the target for inhibition
Answer: Thank you for your kind advice As you mentioned, we carefully checked the Figure 7. According to the results of Figure 4C, KMU-1170 pretreatment suppressed LPS-induced nuclear translocation of NF-κB p65 in THP-1 cells. Therefore, we corrected the inhibitory line going from KMU-1170 to the nucleus. You can see the corrected descriptions in the revised Figure 7.
Minor points:
Line 15: which are
Line 58: essential
Line 230: Protein kinases are enzymes that
Line 239: indolin-2-one
Line 474: one-way ANOVA
Answer: Thank you for your kind advice. As you mentioned, we carefully corrected some typos. You can see the corrected descriptions in the revised manuscript.
Reviewer 2 Report
The paper is very interesting and well written.
The only thing I would recomend is to perhaps disuss the effects the inhibition of the described inflammatory pathway could have.
Also, given the early stage of your easerch, I would recomend you stressed more the need for further studies.
Author Response
#2 reviewer :
The paper is very interesting and well written.
The only thing I would recommend is to perhaps discuss the effects the inhibition of the described inflammatory pathway could have.
Also, given the early stage of your research, I would recommend you stressed more the need for further studies.
Answer: Thank you for your kind advice. As you mentioned, we added what the reviewer recommended. You can see the modified descriptions in the revised manuscript (page 12, line 314).
Round 2
Reviewer 1 Report
The manuscript has been improved according to recommendations.
Author Response
#1 reviewer :
Comments and Suggestions for Authors
The manuscript has been improved according to recommendations.
Answer: Thank you for your kind reviewing of our manuscript.